# Approach for Cost Functions for the Use in Trade-Off Investigations Assessing the Environmental Impact of a Future Energy Efficient European Aviation

Peter Förster [1,*,†] , Bekir Yildiz [1,2,†] , Thomas Feuerle [1,2] and Peter Hecker [1,2]

1   Institute of Flight Guidance, Technische Universität Braunschweig, 38108 Braunschweig, Germany; b.yildiz@tu-braunschweig.de (B.Y.); t.feuerle@tu-braunschweig.de (T.F.); p.hecker@tu-braunschweig.de (P.H.)
2   Cluster of Excellence SE²A—Sustainable and Energy-Efficient Aviation, Technische Universität Braunschweig, 38108 Braunschweig, Germany
*   Correspondence: peter.foerster@tu-braunschweig.de; Tel.: +49-531-391-9862
†   These authors contributed equally to this work.

**Abstract:** Aircraft emissions represent a relevant amount of human induced $CO_2$. Globally, up to 2.5 per cent of such emissions stem from the aviation industry. In order to investigate the effects within the atmosphere, realistic flight profiles are necessary to provide quantitatively tangible values of emissions. The flight profiles and the according fuel consumption can be calculated by using waypoints from flight plans and Base of Aircraft Data (BADA). This paper presents an approach to refine the fuel consumption by integrating the passenger load into the calculation. Since effects of emissions have to be assessed on a greater scale, such as on the European air traffic network, the presented approach provides cost functions for $CO_2$ emissions for different aircraft types and load factors. The cost functions were derived by means of regression analyses of BADA based calculated flight profiles with a step size of one second. The calculations are based on real historic traffic scenarios over several days. The derived aircraft specific fuel burn coefficients enable a simple and efficient integration of $CO_2$ estimations depending on the flight distance, load factor and aircraft type. This can be applied to large traffic scenarios to also study different set-ups such as travel restrictions, other disruptions or an alteration in the traffic system as a whole. In order to enable the assessment of further aspects of such changes to the European air traffic system at large and to foster reproducibility and comparability of related studies, we provide further general-purpose cost estimation functions for several important key characteristics. Besides fuel consumption, we develop cost estimations for air navigation fees and maintenance for conventional aircraft. Those functions are also provided for the design concept of a short-range all-electric aircraft. This propeller aircraft features game-changing technologies such as active laminar flow control, active load alleviation and advanced materials and structure concepts. The approaches discussed in this paper will focus on the generic aspects of aircraft related costs, which can be derived from general available data. For the sake of reproducibility, the results will be made publicly available.

**Keywords:** cost estimation; European air traffic network; simulation

## 1. Introduction

The reduction of $CO_2$ emissions within the aviation industry is a significant goal that is pursued on a global scale. Besides improvements of current propulsion technologies and the progress achieved in aerodynamic facets, other technologies are being investigated as well with great effort. That addresses alternatives to the carbon based aviation, as well as radical new shapes of the aircraft cell. In line with the new propulsion technologies, changes in the flight performance of the aircraft are to be expected. With respect to the current models, climb rates as well as ranges of new aircraft will change. Furthermore, new

propulsion units can contribute to less noise emissions, which can lead to different flight corridors or adapted night flight regulations.

Figure 1 shows the SE²A (the Cluster of Excellence SE²A—Sustainable and Energy Efficient Aviation—investigates future aircraft types, by following an interdisciplinary approach) short range (SR) aircraft with foldable wings and a high aspect ratio. The SE²A SR version 1 (SRV1) features an all-electric propulsion system with two fuselage-mounted electric motor-driven propellers. Concerning the battery energy, a cell energy density of 800 Wh/kg was assumed to be reached by the year 2050 [1].

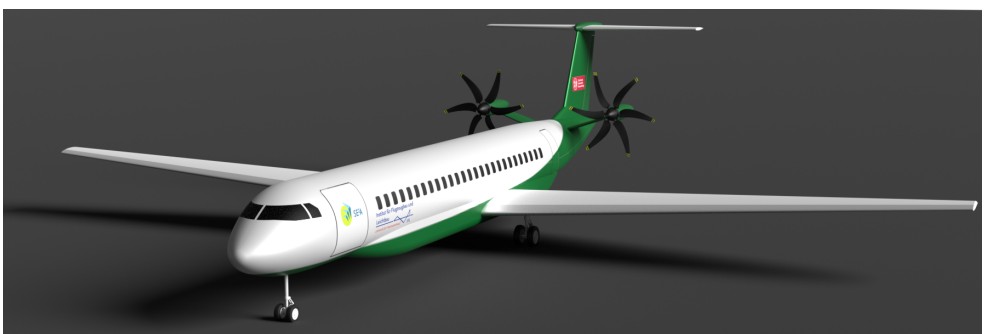

**Figure 1.** SE²A short range all-electric aircraft [2].

One field of research is engaged in the modeling and simulation of the future air traffic system which incorporates those new types of aircraft. The aim is to understand the implications with respect to current operations, which result from introducing the new aircraft types into the present air traffic system. This addresses areas such as the regulation of the aircraft flow or the necessary infrastructure to be provided on the ground. Especially on the ground, substantial changes in the involved processes are to be expected. An overview of the possible new aircraft configurations can be found in [3], where possible implications on the boarding process are investigated for the case of a blended wing body. The expected reduced ranges of electrically operated aircraft might lead to a different structure of the transport flow within the European Air Traffic System (ATS). That is, typical pairs of origin and destination operated by aircraft with distinct passenger capacities might not be available anymore when using electrically operated propulsion systems. Depending on the demand of passengers at each airport, the established flight connections of the ATS might have to be adapted to the new aircraft types that are introduced into the current model range. One aspect of modeling the future air traffic system is to assess the trade-off between reduced ranges and the benefit of saving emissions. To provide tangible results with respect to the atmospheric impact of conventional aircraft types, the simulation has to provide realistic emission values. Due to the variety of involved stakeholders and their multifaceted interactions, as well as the large amount of different operational procedures within the ATS, the complexity of the system generally poses a challenge with respect to computational costs. Therefore the calculations of emissions have to be efficient with respect to the computing demand.

To meet the above mentioned requirements, cost functions were derived. Those estimation functions aim to enable a spatially larger approach to the investigation of atmospheric impacts of conventional aircraft in an efficient fashion.

Besides emissions, this paper aims to provide a selection of estimation functions which need little computational effort to support the assessment of the system from a flow perspective. Detailed processes, such as new procedures on the ground or at the arrival phase are thereby neglected. The estimation functions aim to help related studies that compare different percentages of new aircraft types which are integrated into the ATS with regard to costs.

This study will present three aircraft related cost estimation functions addressing the fuel consumption (with the $CO_2$ emission, respectively), the air navigation charges and maintenance costs. The navigation charges will be presented for all pairs of the selected European airports. The three Direct Operating Cost (DOC) estimation functions will be generated for 9 different aircraft types which constitute the most prominent with regard to the European air traffic.

In Figure 2, Eurocontrol [4] provides an overview of an airline cost structure, covering data derived from 61 airlines. As shown in Figure 2, a variety of costs addresses the business model of an airline. Those are Indirect Operating Costs (IOC), such as the costs for sales and ticketing. The direct operating costs addressed in this paper are marked in blue and can be derived from general available data, such as the aircraft specifications or fees of air navigation service providers. The aim is to provide estimation functions which can be applied in various contexts with little computational effort and which are related to the particular characteristics of the aircraft type or the seat load factor. In a possible future step, other cost aspects of Figure 2, such as airport charges or specific airline business models, could be implemented additionally. Since IOC are not freely available, they will be omitted in this paper.

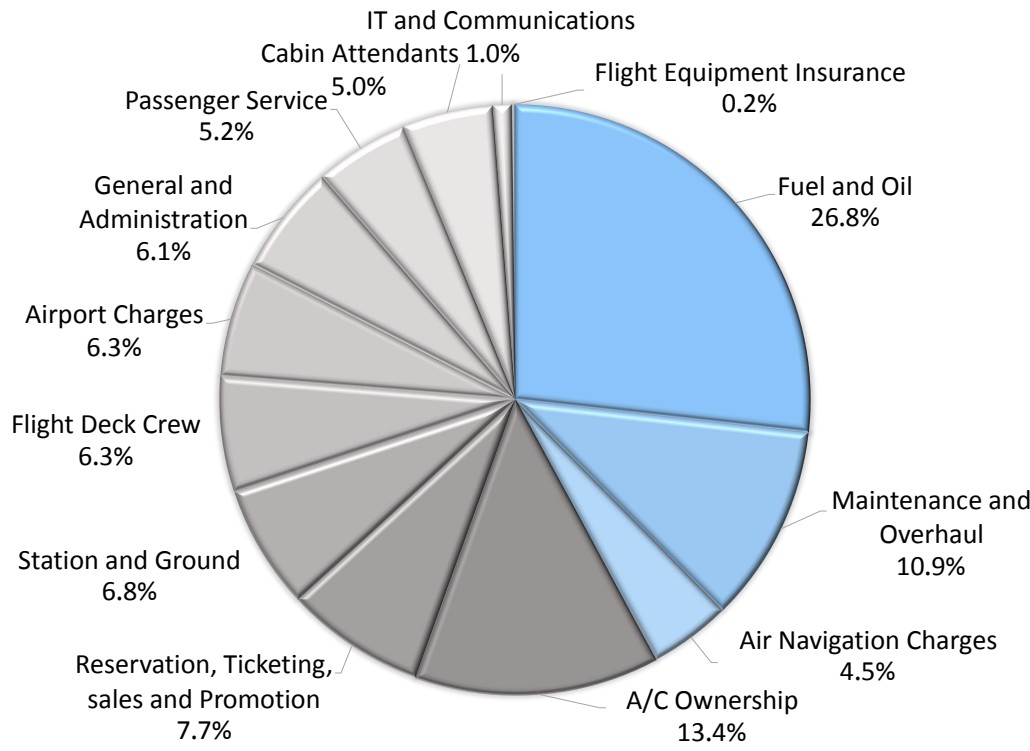

**Figure 2.** Airline cost structure of conventional aircraft, own representation based on ECTL [4].

For one selected new aircraft type, which is electrically powered, the three cost estimation functions will be obtained as well. The functions are developed from free available data and will be composed and published in order to provide a foundation for modelling and simulation from a flow perspective of the future air traffic system facilitating interdisciplinary research. The following paragraph will shortly address the flow problem of a changed network as a motivation for the estimation functions in more detail.

Besides the simulation of the ATS under the conditions of future propulsions system, the question of an optimized traffic flow for different percentages of new aircraft types is one driver for the generation of cost estimation functions presented in this paper. Optimized flow solutions are investigated with respect to minimizing the delay of the air traffic flow management (ATFM), which considers en-route sector as well as airport capacity and focuses on current aircraft configurations, see [5–7]. As mentioned above, this might lead

to a different shape of the system with respect to the present-day situation concerning the airport utilization and the set of connections among the European airports.

Optimization models have to consider the adapted ranges of the new aircraft types as well as airport capacity restrictions, in order to provide a new cost optimized set of connections among the European airports. Due to the complexity of the air traffic system, even by neglecting current economic realities or legal circumstances, it is crucial to produce computational efficient cost functions to facilitate the search for an optimum. In an optimization model, costs can also serve as constraint variables.

In [8], distance and weight depending cost functions for different aircraft types are presented. To enable a better estimation of emitted $CO_2$, this paper will extend this approach by proposing a calculation of the fuel consumption, which includes the particular Seat Load Factor (SLF), thus addressing the varying passenger demand. Since the capacity of a particular airport plays a pivotal role with regard to simulation and optimization problems of the ATS, the paper proposes an estimation for the capacity of all European airports based on the runway layout.

The paper is structured as follows. In Section 2, the data foundation for the computation of the estimation functions is discussed. Section 3 presents the assumption of airports capacity based on the available runway layout. Section 4 shows results of the fuel estimation function based on different load factors, which was derived by a regression of a variety of flight profiles. In Sections 5 and 6, the computation of navigation charges and a selection of maintenance costs are presented. Section 7 applies the approaches of Sections 4–6 to the new all-electric aircraft SRV1. In Section 8, an outlook for possible future applications is given.

## 2. Data Foundation

The cost function of $CO_2$ emissions was founded on calculations of realistic flight profiles, using the Base of Aircraft Data (BADA) data set for specific types of aircraft [9].

The use of BADA was deemed appropriate since recorded real life flight data of different aircraft types and routes were not available in the required amount or detail. Since the paper aims to improve the fuel calculation by considering different levels of utilization, that is the seat load factor SLF, which relates the number of transported passengers to the seat capacity of the aircraft, a quantitative tangible implementation of fuel burn was selected. The widely used, simplified point mass based BADA family 3 framework, which covers a large variety of aircraft types was used for this purpose. It provides quantitatively sufficient results with respect to thrust and related fuel burn, which itself constitutes the foundation for $CO_2$ estimations.

The calculation of thrust and fuel burn of different phases of flight, such as taxiing, take-of, climb, cruise or descent are estimated by equations which are founded on the BADA point mass model, which incorporates particular aircraft related coefficients. The implementation of BADA used for this study detects the transition of flight phases and applies the according calculation. In order to better reflect real flight trajectories, turning flights were implemented as well. For each type of aircraft, a variety of flight profiles were used to calculate the respective fuel burn.

The profiles were construed by using given waypoints which are part of the available flight plan data. Weather influences were neglected and the standard atmosphere was applied for all BADA calculations.

The calculation of vertical and horizontal phases of the trajectory incorporates different aspects such as rate of climb or descent (ROCD), true airspeed $V_{TAS}$ and fuel consumption which relates to the change of mass of the aircraft. With regard to the implementation of the turning flight, the lateral flight profile provides an adequate horizontal path defined by waypoints and legs as well.

As shown in Figure 3, in the trajectory calculation, the transition from one nominal straight line to the other is simplified by a circular arc. Coming from the waypoint $WP_{i-1}$, if the aircraft reaches the switching point $P_{zi-1}$, then the next waypoint $WP_{i+1}$ of the flight

plan is targeted and the aircraft begins a turn to the next course prior reaching this waypoint. The turn arc or the curved path segment is centered in $M_i$ with radius $r_i$. The rate of turn $\dot{\chi}$ can be calculated as follows:

$$\dot{\chi} = \frac{g_0}{V_{TAS}} \cdot \tan(\theta) \qquad (1)$$

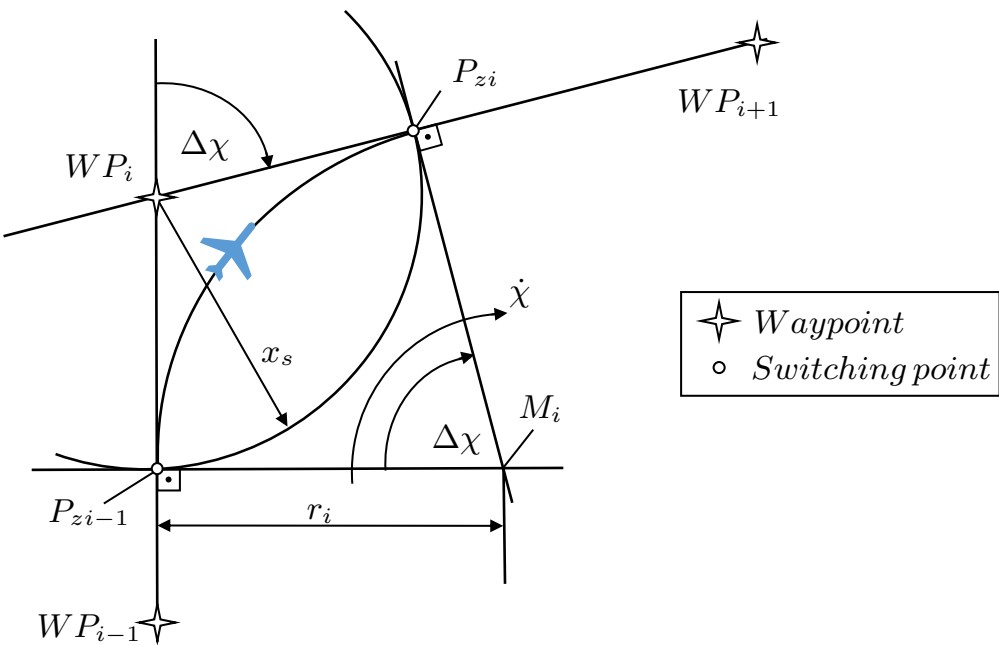

**Figure 3.** Turn anticipation and geometry of the circular arc.

Here, $g_0$ is the gravitational acceleration. The control input, bank angle $\theta$, is based upon the amount of course angle change $\Delta\chi$ between the two flight legs and ranges between 15° and 35°. The distance $x_s$ between the initial switching point and the so called fly-by waypoint $WP_i$ is ensured by Expression (2) [10]. Thus, the switching point $P_{zi-1}$ can be set, if $WP_i$ is known.

$$x_s = r_i \tan\left|\frac{\Delta\chi}{2}\right| = \frac{V_{TAS}}{\dot{\chi}} \tan\left|\frac{\Delta\chi}{2}\right| \qquad (2)$$

After the second switching point $P_{zi}$ or the target course is reached, the turn is completed and the aircraft flies along the great circle path until the next waypoint of the flight plan is targeted. This procedure is repeated for all waypoints separating the two route segments. The vertical flight profile is construed by the conjunction of several flight segments represented in the initial flight plan.

The waypoints from the flight plan, which were used to generate the flight mission profiles, were procured from the "so6" data set, provided by [11]. These data encompass all planned flights within Europe, including pairs of origin and destination, all planned waypoints and the type of aircraft.

Figure 4 shows an example of the implementation of the BADA model. The upper left part depicts a reference flight profile (from yellow to blue) of an Airbus A320 which was generated by a human in the loop simulation (HITL). This flight profile of a short range flight was conducted by means of the cockpit simulator of the Institute of Flight Guidance of TU Braunschweig (see Figure 5 upper right part). This simulator incorporates the Prepar3D V3 Engine. The reference scenario represents all relevant flight phases. The upper right part of Figure 4 shows the according course of true airspeed, TAS, and altitude.

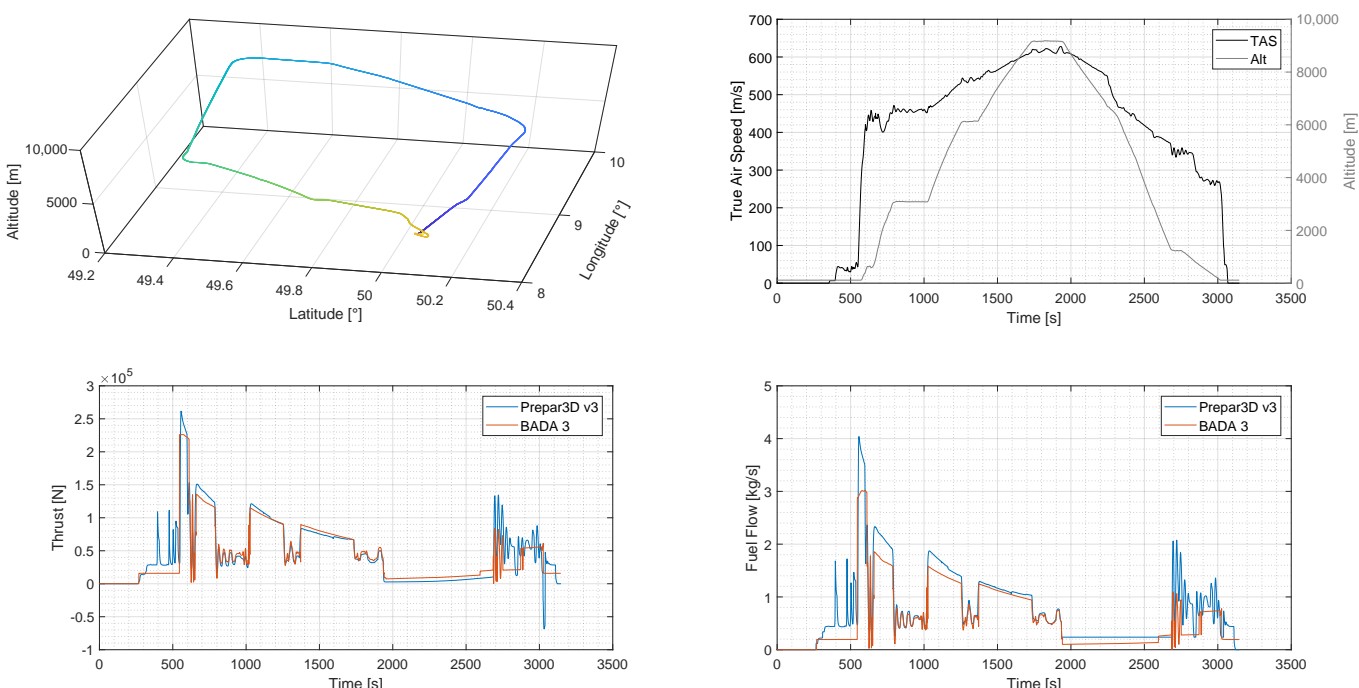

**Figure 4.** Verification of the BADA implementation.

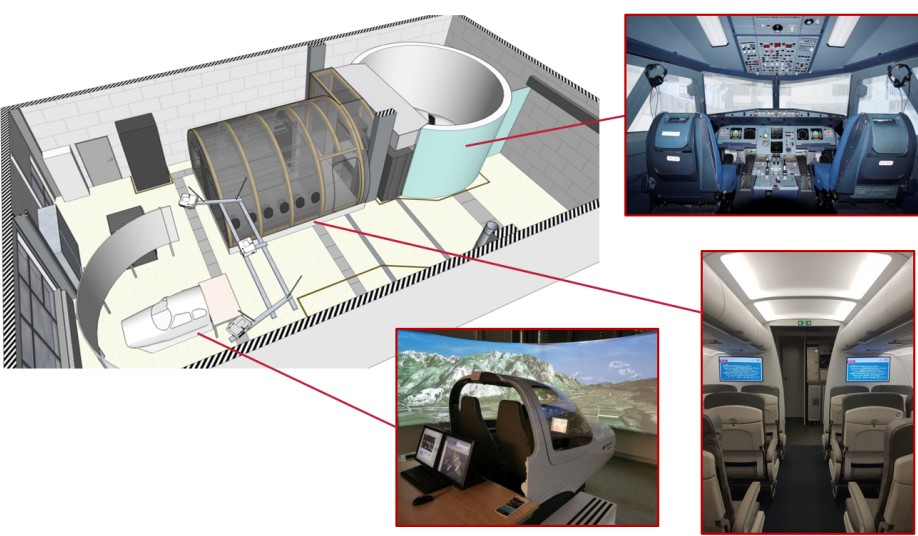

**Figure 5.** Flight simulators of the Institute of Flight Guidance of TU Braunschweig.

In the lower left part of Figure 4, the input of thrust from the human operator is represented as the blue line. The implemented BADA model detects the distinct phases of flight and applies the respective thrust configurations. For example, during the taxi phase between approx. 280 s and 520 s, different inputs from the human operator can be observed. This is common procedure during that phase. The estimation of the BADA model for this phase on the other hand assumes a constant thrust. Varying human inputs can also be observed during the landing phase between approx. 2800 s and 3000 s.

Again, the BADA approximation assumes constant thrust values for this phase. This leads to differences in thrust and the related fuel consumption, as illustrated in the lower right part of Figure 4. Furthermore, the differences between HITL and BADA in fuel flow during a climb segment between 600 s and 800 s are greater than in the corresponding

thrust values. This can be assigned to the influence of the particular BADA coefficients. The fuel flow itself constitutes a function with the following dependencies (see [9]):

$$fuelflow = f(thrust, flightphase, TAS, coefficients_{BADA})$$

The thrust calculation of BADA for lateral flight phases such as between 800 s and 1000 s was extended by considering instationary flight states as well. In the upper right part of Figure 4, instationary flight states are represented for example by an oscillating TAS. The lower left and lower right part of Figure 4 show a good matching between HITL generated values and the BADA implementation. In the presented example of an Airbus A320, the difference between HITL and BADA data during the cruise phase between approx. 1700 s and 1900 s follows to 4.347%. It should be stressed, that the majority of flight kilometers is related to lateral flight phases.

In the example of the available data set of 122 days used in this paper, 62.67 percent of kilometers fall into this category. This applies for the months March, June, September and December of the year 2018. For the purposes of the $CO_2$ estimation in this paper, we deem the BADA approach for generating flight profile data as sufficient precise enough.

The data used for generating the flight profiles also represent international incoming and outgoing traffic. Besides the calculation of $CO_2$ emission, the "so6" data set was used to limit the amount of airports to the European flight network. This is due to the fact that the data set of other regions only contains European related flights. Since the European airspace is heavily connected with other regions in the world, a calculation of the interregional traffic flows was performed, in order to assess the amount of traffic related to other regions.

In that respect, the following paragraph will discuss the so6 data foundation in more detail. For the purposes of illustration, the 25 August 2018 will serve as an example of the air traffic situation across Europe. Figure 6a,b visualize the traffic flow over Europe, derived from the so6 data set considering updated waypoints. Both figures represent a Mercator projection. Figure 6a depicts a cutout of Europe with all 1216 airports, airfield or oil rigs and the waypoints filed for the planned data set. The underlying sample data embodies 33,674 flights and approximately 2.6 million waypoints. Figure 6b aims to reveal the traffic density. Here, due to irregular distributed waypoints, with respect to flight profiles, an interpolation was carried out. To better visualize the traffic streams and focus areas from Figure 6a, a resolution of one second was chosen for interpolation. This procedure leads to approximately 324 million data points, illustrated in Figure 6b. The heat map clearly illuminates the focal areas of flight movements in Europe. It illustrates qualitatively the number of (interpolated) waypoints per area. The main streams of flow are also highlighted. Both figures illustrate the complexity of the European air traffic. The following paragraph elaborates on the process of excluding airports in the context of traffic streams across different regions.

Figure 7 depicts the international regions of air traffic, as classified by ICAO [12]. Since this paper aims to provide generic cost functions for the European air traffic, the regions E and L have been specially investigated. In the process of selecting relevant airports, firstly, oil rigs and airfields with no commercial air traffic were removed from the data. Secondly, a calculation of the traffic streams between the affected ICAO regions was carried out.

The data show, that up to approximately 73% of all flights, listed in the so6 records take place within the regions E and L. We deem the regions E and L as sufficiently representative in the light of related European traffic streams. E and L will therefore define the system boundaries, though the influx of other regions, which will be excluded in the following considerations, is acknowledged.

Figure 8a,b illustrate the traffic flow among the different ICAO regions with respect to the European airspace for selected days. As one aspect, the differences between the particular summer and winter season are revealed in this depiction. For example, the flow between U and L in Figure 8a decreases by up to 48% between the month of July and January. Due to missing data of particular months of a year, the calculation illustrated in

Figure 8a,b were performed by means of data of different years. This is because so6 data became unavailable for public access and not all data could have been secured.

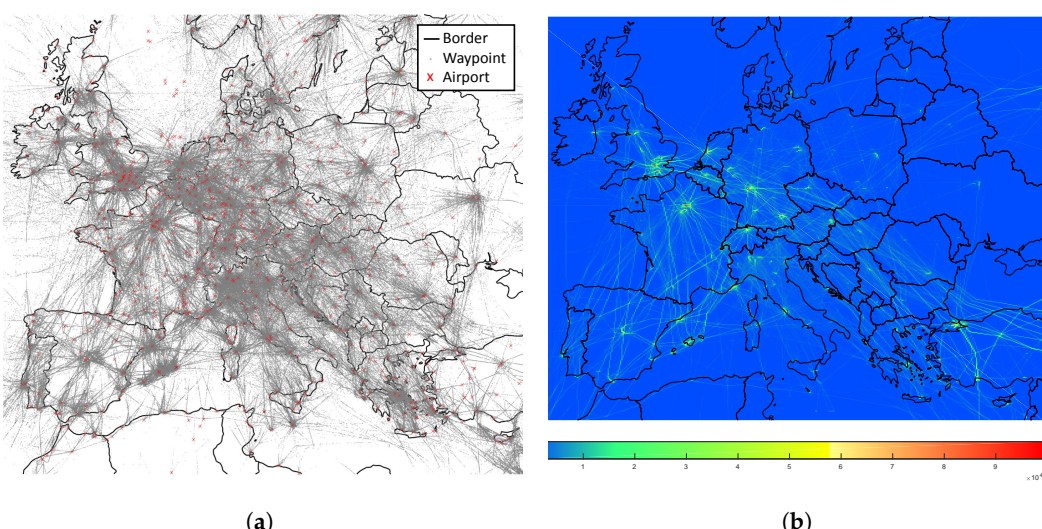

(**a**)　　　　　　　　　　　　　(**b**)

**Figure 6.** Air traffic situation across Europe. (**a**) Cutout of European airports, airfields and oil rigs and related waypoints of the so6-M3 data set of the 25 August 2018. (**b**) Corresponding heat map of the data set , using interpolated waypoints.

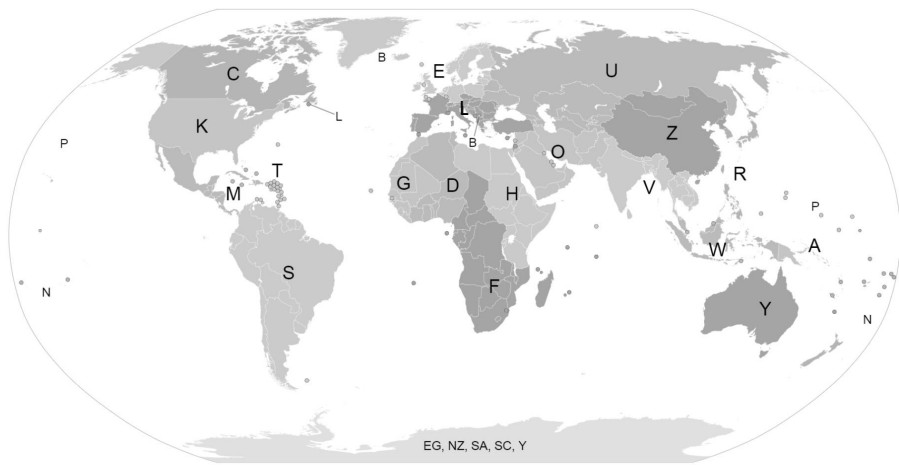

**Figure 7.** Regions according to ICAO classification, adapted from [12].

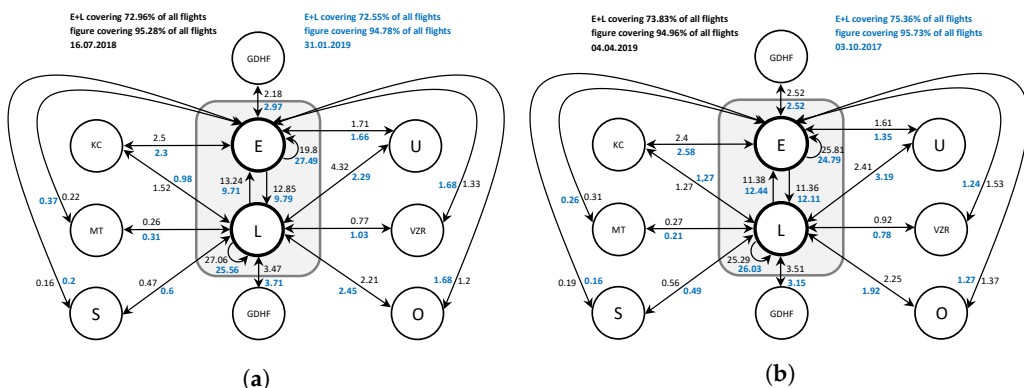

(**a**)　　　　　　　　　　　　　(**b**)

**Figure 8.** Percentages of traffic flow for the regions E and L of the ICAO classification, calculated for 4 different seasons. (**a**) Traffic flow July 2018 and January 2019. (**b**) Traffic flow April 2019 and October 2017.

### 3. Calculation of Airport Capacity

With respect to flow based simulations of the European air traffic or optimization problems, the airport capacity plays a pivotal role.

This chapter adapts an estimation of airport capacity which is based on the runway configuration and was developed by the Federal Aviation Administration, see [13]. The approach of estimating the runway capacity was applied on European airports.

Generally, the capacity of an airport depends upon several factors, such as the number and layout of runways, the aircraft mix or operational aspects like the separation between aircraft that are imposed by the air traffic authorities as a result of the current demand, weather influences or noise constraints [14]. As a good rule of thumb to estimate the Declared Runway Capacity (DRC), the 5% peak hour movement is often applied, since it is expected that the DRC reaches at least the maximum number of aircraft movements per hour [14]. In their study of the capacity utilization of 75 European airports, Schinwald and Hornung [14] revealed that most airports only use low to moderate capacity and only seven operate at their highest capacity (EGLL, EDDF, LTBA, LIML, LEMD, EDDM, LFPG).

This paper aims to apply an approach to automatically determine the runway layout category by means of available topological data in order to use an approach to calculate the runway capacity.

The latter approach incorporates the number and layout of the runway as well as the particular mix of traffic. In this approach, the capacity of an airport is considered as the annual and hourly capacity. It is derived from the particular runway layout of an airport, which follows the methodology of [13], where an analysis of the annual service volume (in operations per year), was made for different combinations of fleet mixes, runway configurations and operational directions.

Since the average yearly fleet mix can be derived from the European so6 data set, the types of different runway configurations, which are assigned with a particular capacity, have to be identified. The fleet mix index is given be the following Equation [13].

$$MI = C + 3D \tag{3}$$

C and D represent the percentage of aircraft of a distinct weight class of the total aircraft mix. Appendix A provides the weight classes used in the method. It should be noted, that since the classes A and B in Table A1 are not used in Equation (3), they are a subset of the total aircraft mix.

Different operational directions of runways show similar or equal values. They were therefore summarized. The runway configurations were identified by means of a formal description of their relative position to each other. The necessary layout plans of each runway were procured from a freely available data set [15], which provides specific airport data. The calculation algorithm proposed in this paper covers most of the cases, that are represented in [13] and enables an automatic classification of airports. The capacities of the particular runway configurations, which are not listed, were assigned manually. The fleet mix *MI* is calculated for an annual time horizon based on the available so6 scenario data. Hereinafter, a formal description of the algorithm will be presented, which was used for the assignment of a particular runway layout which was then used to classify the respective capacity.

The available layout data of the individual airports (see [15]) provide start and end points $(s, e)$ of the particular runways. In order to process conformal Cartesian values of those points, the provided latitude $\varphi$ and longitude $\lambda$ values are transformed into a Mercator image by $X = \lambda a_{WGS84}$ with $a_{WGS84} = 6378.137$ km and

$$Y = a_{WGS84} \cdot \ln\left[\tan\left(45° + \frac{\varphi}{2}\right)\left(\frac{1 - e\sin\varphi}{1 + e\sin\varphi}\right)^{\varphi/2}\right] \tag{4}$$

with

$$e^2 = \frac{a_{WGS84}^2 - b_{WGS84}^2}{a_{WGS84}^2}, \ b_{WGS84} = 6356.752 \text{ km}$$

respectively. The calculated Cartesian values are than arranged as vectors, leading to the start $\vec{r}_{m_s}$ and endpoints $\vec{r}_{m_e}$ of a runway $m$:

$$\vec{r}_{m_s} = \begin{pmatrix} r_{m_{sx}} \\ r_{m_{sy}} \end{pmatrix}, \vec{r}_{m_e} = \begin{pmatrix} r_{m_{ex}} \\ r_{m_{ey}} \end{pmatrix} \text{ and } \vec{r}_m = \vec{r}_{m_e} - \vec{r}_{m_s} \tag{5}$$

In order to derive the particular type of a runway system, in comparison with the provided scheme by [13], the runways have to be examined with regards to parallelism, in between distances and possible intersections. Deducting if two runways $m, n$ are parallel leads to:

$$\alpha_{r_m r_n} = \arccos\left[\frac{\vec{r}_m \cdot \vec{r}_n}{|\vec{r}_m||\vec{r}_n|}\right], \text{with} \ \alpha_{r_m r_n} \leq \varepsilon \tag{6}$$

with a given error bound $\varepsilon$. The distance $d$ between two parallel runways $m, n$ can be derived from the distance of both starting points $\vec{r}_{m_s}, \vec{r}_{n_s}$:

$$d = \frac{|\vec{r}_m \times (\vec{r}_{n_s} - \vec{r}_{m_s})|}{|\vec{r}_m|} \tag{7}$$

Lastly, in the case of nonparallel runways, possible intersections are calculated by solving the linear system, formed by equating both linear equations of $\vec{r}_m$ and $\vec{r}_n$:

$$a\frac{\vec{r}_m}{|\vec{r}_m|} - b\frac{\vec{r}_n}{|\vec{r}_n|} = \vec{r}_{n_s} - \vec{r}_{m_s}, \text{or} \underbrace{\begin{pmatrix} \frac{r_{m_x}}{|\vec{r}_m|} & \frac{-r_{n_x}}{|\vec{r}_n|} \\ \frac{r_{m_y}}{|\vec{r}_m|} & \frac{-r_{n_y}}{|\vec{r}_n|} \end{pmatrix}}_{R} \underbrace{\begin{pmatrix} a \\ b \end{pmatrix}}_{\vec{x}} = \underbrace{\begin{pmatrix} r_{n_{sx}} - r_{m_{sx}} \\ r_{n_{sy}} - r_{m_{sy}} \end{pmatrix}}_{\vec{r}_s} \tag{8}$$

and thus

$$\vec{x} = R^{-1}\vec{r}_s \tag{9}$$

with $0 \leq a, b \leq |\vec{r}_m|, |\vec{r}_n|$ in case of an intersection.

With the formal representation given above, we deducted the airport runway capacity based on the specific runway data of an airport given in [15] and the yearly fleet mix, which can be calculated using the so6 data that represent all aircraft using a particular airport. With regards to the classification available in [13], a couple of simplifications were made. Table A4 gives an overview of the different runway types with respect to the annual and yearly capacity. The cases of G & H and I & J were summarized for this calculation.

The complete list of the assigned airport capacity for all European airports can be found under https://bitbucket.org/bekir114/input-data/downloads (accessed on 24 January 2022).

Figure 9 depicts an example of European airports, which do not fit into the existing Runway Configuration (RWC). For the sake of transparency, the depiction in Figure 9 does not show a uniform scale.

As an example of the applied algorithm, two airports, Munich and London were investigated. Given the available flight plan data for the year 2018, the fleet mix index follows to 116.2 and 178.4, respectively. The 95th percentile value of the traffic volumes for the hourly movements are calculated to 74 and 83, respectively. This fits well with the data in Appendix C. Munich and London represent the runway configuration case C.

The difference between the deducted values and Appendix C can be assigned to the data foundation. Since the calculated numbers are based on flight plan data, it is assumed that real life data , due to weather or other influences, will differ in many cases. Thus, showing a higher number in hourly movements. To better assess the runway capacity, it would be necessary to process real life recorded data, such as Automatic Dependent

Surveillance-Broadcast, ADS-B. Since this paper applies the approach of [13] and focuses on the assignment of runway layout, this fell out of scope.

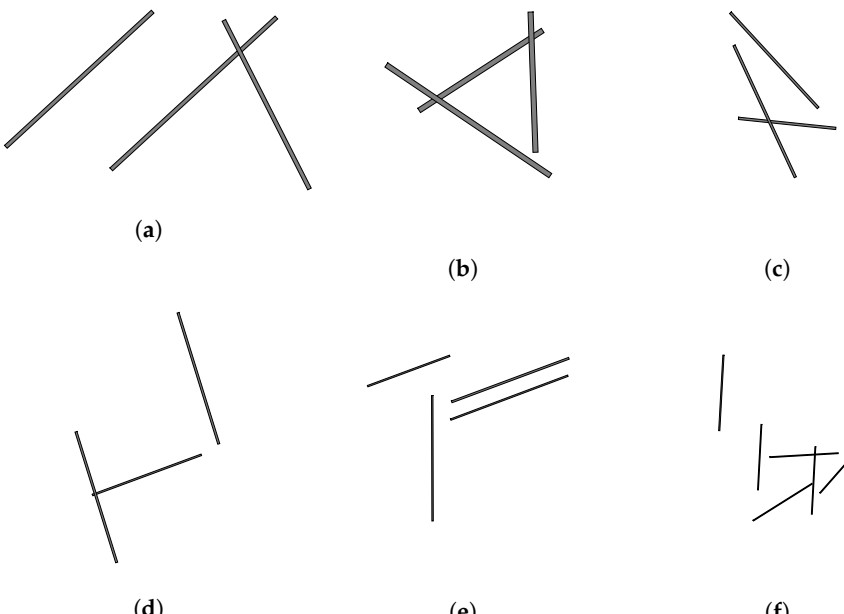

**Figure 9.** Examples of exceptional runway configuration layouts, generated by the algorithm with relation to the Horonjeff classification. (**a**) Helsinki-Vantaa ICAO: EFHK, RWC: **C** . (**b**) Chalgrove ICAO: EGLJ, RWC: **H**. (**c**) Zurich ICAO: LSZH, RWC: **M**. (**d**) Rome–Fiumicino ICAO: LIRF, RWC: **L**. (**e**) Frankfurt am Main ICAO: EDDF, RWC: **M**. (**f**) Amsterdam Schiphol ICAO: EHAM, RWC: **E**.

The following chapter will propose an estimation function for the calculation of $CO_2$ costs.

## 4. Estimation Function of $CO_2$ Emissions

The aim of this chapter is to provide a function, which estimates the amount of $CO_2$ emissions, depending on the aircraft type, the flown distance as well as the number of passengers. The available data of the flight plan (see [11]), which reflect the waypoints of a planned trajectory, are used to create this function by means of a regression analysis. It may be noted, that two different types of data, which vary in terms of completeness were used in this approach.

As mentioned above, the first type of data comprises planned trajectories (so6-M1). The second part is called so6-M3. They represent an updated version of so6-M1, which is based on radar data and covers the flown trajectories.

For the sake of general application, the 39 most used aircraft types were considered in the estimation of fuel consumption. For those aircraft of type $t$, specifications of emissions $e_t$, range $r_t$ and seat capacity $s_t$ were considered. Taking into account the available flight plan of the selected time period, there are more than 300 different types of aircraft available.

Nevertheless, for many less frequent aircraft types, the flight plan does not provide enough data to perform a regression analysis. The regression which will be proposed in this chapter is performed on the basis of 100 different routes $R_m$, $m$ = 1:100. Since only the two most prominent aircraft types of Airbus A320 and Boeing 738, with a coverage of 39.7% of all flights, provide a large amount of different routes, we decided to limit the number of routes to 100. Figure 10 illustrates the distribution of different flight routes per aircraft type for the reference day of 25 August 2018. The selected 39 most used aircraft types represent 83.06% of all European flights.

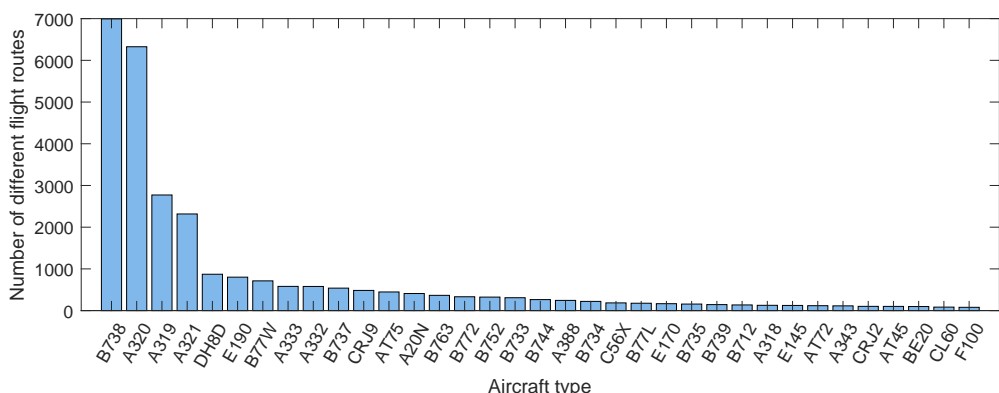

**Figure 10.** Cutout of the number of flight routes related to aircraft types, 25 August 2018.

The amount of emissions, in our case $CO_2$, depends on the fuel burnt during a flight, the type of engine, as well as on the phase of flight (see ICAO data [16]). The number of passengers contributes to the fuel consumption, because a higher mass relates to a higher rate of fuel burn $\dot{m}_{fuel}$. A lower starting mass, which declines during the course of a flight, results in a lower amount of fuel burnt during this flight leg. Being a function of distance and mass, the optimum of fuel economy for an Airbus A320 for example ranges between 1500 km and 5000 km [17].

As a first simplified assumption for the amount of $CO_2$ emissions, we apply the commonly known emission index

$$\text{EI}_{CO_2} = 3.15 \frac{\text{kg}_{CO_2}}{\text{kg}_{fuel_{burnt}}}. \tag{10}$$

This factor is widely used for estimations of $CO_2$ emissions, such as in the SESAR Performance framework, where it serves to calculate relevant Key Performance Indicators (KPI) (see [18]). Other applications of the factor can be found in trade-off estimations with respect to optimized trajectories for less $CO_2$ emissions at great altitudes (see [19]).

The proposed approach calculates the amount of fuel burnt, depending on the distance of the particular flight leg between two airports $i$ and $j$, $l_{ij}$ and the respective seat load factor, $S_{ijt}$. The latter representing the utilization of the available seats $s_t$ of a particular type of aircraft $t$ at the flight between $i$ and $j$. The amount of fuel burnt can thus be formulated as follows:

$$fuel_{ijt} = f(S_{ijt}, l_{ij}) \tag{11}$$

By means of BADA [9], the rate of fuel burn $\dot{m}_{fuel}$ and thus the amount of fuel used during the flight leg, can be calculated for specific types of aircraft and for particular routes $R_m$. The BADA data embody the particular aircraft performance envelopes, whereas the so6 data set provides scenarios of realistic flight profiles. The amount of emissions for a particular connection will therefore be calculated by

$$e_{ijt} = EI_{CO_2} \cdot fuel_{ijt}[kg] \tag{12}$$

As discussed above, the calculation of $fuel_{ijt}$ is governed by the mass of the aircraft, which also includes the mass of fuel necessary to reach a particular distance, $m_{tripfuel}$. The following equation reflects the particular components of the mass $m$ of an aircraft of type $t$.

$$m = m_{OEW} + m_{pax} + m_{cargo} + m_{tripfuel} \leq MTOW \tag{13}$$

with *MTOW* denoting the maximum take-off weight and *OEW* denoting the operating empty weight. In the proposed approach, the portion of cargo will be neglected, and we

assume the average mass of a passenger with associated luggage to be 100 kg. Given a starting mass of

$$m_{start} = m_{OEW} + m_{pax} + m_{tripfuel_{start}} \qquad (14)$$

with $m_{start} = m_{ref}$ ($m_{ref}$ as taken from BADA), the calculation of $\dot{m}_{fuel}$ for a particular flight leg is carried out iteratively.

For a given Route $R_m$ and number of passengers, (expressed by the seat load factor $S_{ijt}$), the necessary fuel $m_{tripfuel}$ can be obtained by calculating the flight legs by means of the implemented BADA model, $f_{BADA}$. The model $f_{BADA}$ computes the range $r_n$ at simulation step $n$, which is obtained by the given starting mass $m_{start_n}$ under the conditions of the given waypoints and the according procedures during the different phases of flight. The latter for example addressing the thrust during climb phases.

The range $r$ that can be reached can be expressed with the placeholder for the implemented BADA function, with $R_{l_{ij}}$ representing a route between the airports $i$ and $j$ with the lateral distance $l_{ij}$.

$$r_n = f_{BADA}(m_{start}^n, S_{ijt}, R_{l_{ij}}) \qquad (15)$$

$m_{start}^n$ is changed at the next iteration step n + 1 to match the given distance $l_{ij}$. As long as $r_{n+1} \neq l_{ij}$ holds, the procedure is being repeated.

In order to match $m_{tripfuel}$ with the given distance $l_{ij}$ or $r$, respectively, we decided to abort the calculation when the difference of mass of fuel falls below 10 kg:

$$|m_{start}^{n+1} - m_{start}^n| < 10kg \qquad (16)$$

Since $m_{OEW} + m_{pax}$ is known, $m_{tripfuel}$ can also be expressed in terms of the route and seat load factor

$$m_{tripfuel}^{n+1} = f_{BADA}(m_{tripfuel}^n, S_{ijt}, R_{l_{ij}}) \qquad (17)$$

It has to be stressed that since different flight routes might resemble concerning the distance $l_{ij}$, for example: $l_{ij}(R_m) = l_{ij}(R_{m+1})$, they can show differences in the given waypoints, which can lead to different fuel consumption.

Figure 11 illustrates the differences between the lateral flight path and according distance $l_{ij}$ and vertically deviating waypoints for an Airbus A320.

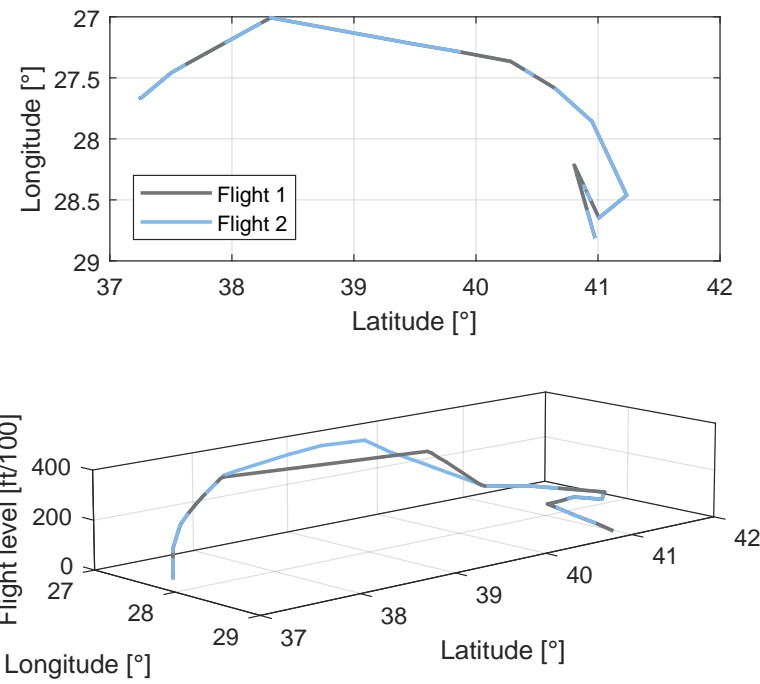

**Figure 11.** Comparison of flight plans for an Airbus A320.

Given the goal is to provide a simple function of $e_{ijt}$, that reduces computational effort when for example applied in an a optimization model, a regression was applied. Therefore, values of $m_{tripfuel}$ were calculated for 100 different flight routes $R_m$ for each of the selected 39 types of aircraft by means of the implemented BADA model $f_{BADA}$. Different seat load factor were chosen to represent a staggered occupancy of 0, 20, ... 100% utilization to show the influences of passengers.

Subsequently, a multivariate regression (linear regression model, using least square method) was applied to the dependent variable $m_{tripfuel}$ and the independent variables $S_{ijt}$ and $l_{ij}$, leading to

$$m_{tripfuel} = C_{1t}l_{ij} + C_{2t}S_{ijt} + C_{3t} \tag{18}$$

In a second step, more detailed operational aspects, with respect to the necessary fuel were considered. This comprises fuel for contingency measures such as mandatory holdings and taxi time, $time_{taxi}$, as well as the consideration of alternate airports (with an assumption of being within a range of 50 nautical miles).

It has to be stressed, that parts of those additional quantities do not represent the overall amount of fuel burnt, since they are only needed for safety issues. On the other hand they increase the mass and therefore the emissions. The contingency measures are commonly reflected by five percent of the necessary trip fuel. The alternate airport can be assumed as a completely new flight leg (given the aircraft does descend to the destination airport and is then informed to navigate to the alternate destination) and thus be considered by a new equation for the trip fuel:

$$m_{tripfuel_{extended}} = (C_{1t}l_{ij} + C_{2t}S_{ijt} + C_{3t})1.05 \\ + (C_{1t}50NM + C_{2t}S_{ijt} + C_{3t}) \tag{19}$$

With the complete mass of trip fuel now available, the amount of fuel burnt, for a particular type of aircraft $t$ was calculated by means of BADA data, analogue to Equation (17):

$$m_{fuelburnijt} = f_{BADA}(m_{tripfuel_{extended}}, S_{ijt}, R_{l_{ij}}) \tag{20}$$

$m_{fuelburnijt}$ refers to the fuel which was actually used up during the flight under normal operations. A second regression analysis leads to:

$$m_{fuelburnijt} = D_{1t}l_{ij} + D_{2t}S_{ijt} + D_{3t} \tag{21}$$

After considering the fuel consumption related to the airborne aircraft, the effect of the taxi time on the ground will be considered. The taxiing speed was hereby set to 15 knots (the necessary thrust represents seven per cent of the maximum thrust). Futhermore, a position at sea level as well as the conditions of the norm atmosphere was assumed. As an applicable value of the average taxi time, $time_{taxi_{avg}} = 26$ min are proposed [20]. In order to reflect the operational circumstances at particular airports better, in a future step, more average taxi times, depending on the specific airport, could be implemented. Considering Equation (21) the taxi time leads to:

$$m_{fuelburnijtaxit} = D_{1t}l_{ij} + D_{2t}S_{ijt} + D_{3t} + D_{4t}time_{taxi} \tag{22}$$

The factor $D_{4t}$ is directly derived from the implemented fuel consumption model for taxiing with the above mentioned assumptions. Equation (22) can now be used in Equation (12) to calculate the emissions during a given distance

$$e_{ijt} = EI_{CO_2}(D_{1t}l_{ij} + D_{2t}S_{ijt} + D_{3t} + D_{4t}time_{taxi}) \tag{23}$$

Figure 12 illustrates the relationship between fuel, distance and seat load factor for an Airbus A340-300, as expressed in Equation (21). The constant factor during the taxi phase as considered in Equation (22) is neglected here, since it is independent from the particular

flight profiles. As mentioned above, every seat load factor for a particular aircraft $S_{ijt}$ is represented by 100 different flight routes. With a coefficient of determination R² of 0.9638, the linear approximation seams to constitute a feasible approach. It has to be noted, that the influence of the seat load factor comes into play at large distance.

Against the expectation of a significant quadratic relationship, due to the influence of short range flights with a higher percentage of airport and climb phases, which result in more fuel consumption, a linear relationship does fit well for the observed real life flight profiles. For example, two aircraft types from the two most used aircraft families in Europe, B737-900 and A321-200, see Figure 13, show a good linear relationship when investigated. In Figure 13, the different seat load factors are summarized, thus leading to a variety of different MTOW values. The differences between quadratic and linear regressions for all seat load factors of both aircraft types result in 0.1095% and 0.3237% for R². For a later application in a network optimization problem, we deem the computational efficient linear approach as adequate and beneficial.

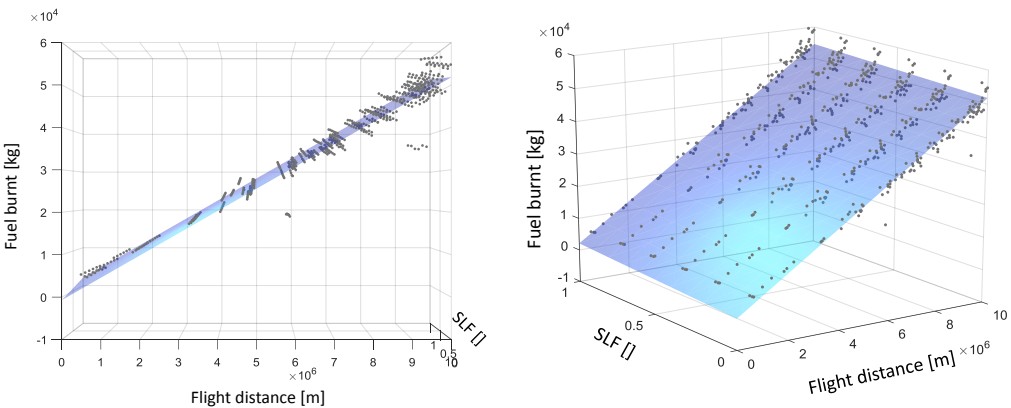

**Figure 12.** Fuel burn regression for an Airbus A340-300 with a coefficient of determination R² = 0.9638.

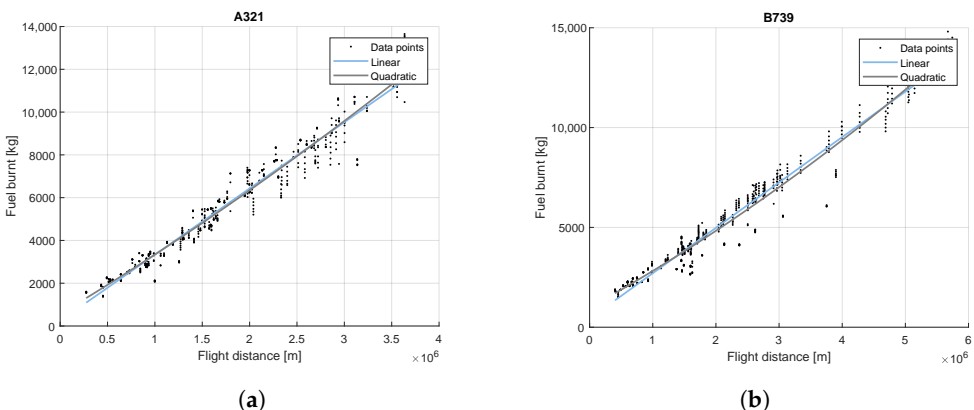

**Figure 13.** Regression analysis of A321 and B739 with coefficients of determination R². (**a**) $R^2_{linear} = 0.9534$, $R^2_{quadratic} = 0.9544$. (**b**) $R^2_{linear} = 0.9684$, $R^2_{quadratic} = 0.9715$.

Appendix B provides the respective coefficients of Equation (22) for 39 types of aircraft. A field of application for the proposed $CO_2$ estimation function could be a flow optimization problem, where under the constraints of a $CO_2$ limit and the European wide passenger demand, an optimal network flow could be determined.

In order to achieve comparability with respect to the regression analysis, the calculation of the flight trajectories and the according fuel consumption was based on the standard atmosphere. Weather related influences such as wind and pressure, or seasonal effects, such as temperature are neglected. Especially for smaller aircraft types, it was necessary to refer to both, summer as well as winter schedule to compile the amount of 100 different flight

routes, which constitute the basis for the regression analysis. As an example of weather effects, the temperature will affect the thrust. Higher temperatures decrease the climb rates which than relate to the fuel consumption. The lateral influence of wind will affect the ground speed accordingly since the True Airspeed, TAS will remain constant. One has to acknowledge that neglected weather influences constitute a bias in the determination of the regression coefficients. For each particular weather condition, a specific set of regression coefficients would have to be calculated. Providing particular weather dependent coefficients was not scope of the study. With respect to wind influences, a simplified approach, in order to avoid a new trajectory calculation with a subsequent regression analysis, would be to convert the differences in ground speed between standard and disturbed atmosphere into a difference in duration. Given the same TAS during disturbed and undisturbed conditions, the simplified estimation of the difference in duration can be translated into distance $l_{ij}$, which itself can be used directly in Equation (23).

## 5. Estimation of Air Navigation Charges

The following two Sections 5 and 6 will attend to the cost aspects of air navigation charges and maintenance and repair. With regard to similar results between the particular aircraft types, the aircraft types in those two chapters will be limited to nine. With respect to the flow optimization problem, we deem 9 different types (Boeing 737-800, Airbus 320, Boeing 777-200, Airbus 330-300, Boeing 747-400, Airbus 340-300, Boeing 767-300, Airbus 380 and Embraer E-190), which represent the most relevant aircraft in the European air traffic system as sufficient. According to Eurocontrol [4], these aircraft types represent the most used ones.

At this point, in contrast to selecting the most used aircraft, a method considering the specific transport capacity, should be addressed briefly. The particular maximal transport capacity can be used as a measure for aggregating different types of aircraft with respect to their frequency within the European air traffic system.

Figure 14 illustrates the relationship between the transport capacity $TC_t$ of a specific aircraft type $t$ and the seat capacity. According to the flight plan of the used data set and a seat load factor of 100%, the transport capacity follows to $TC_t = n \cdot t_{pax_{max}}$, with $n$ as the total number of flights of an aircraft type $t$ and $t_{pax_{max}}$ as the maximum passenger capacity . This is depicted at the right y-axis of Figure 14, where the selected 9 types of aircraft are registered as well.

When tackling the aspect of decreasing the amount of aircraft types, by incorporating the transport capacity $TC_t$, the class of aircraft can be considered as an approximated function, as derived from the diamond marked data points in Figure 14.

$$TC_t = f(t_{pax_{max}}) = f(t) \tag{24}$$

The resulting generic aircraft classes can then be weighted by $TC_t$. Depending on the boundaries $t_b$ and $t_a$ of a new generic class, $class_{ipax_{max}}$, the new maximum number of passengers of a new class would lead to

$$class_{ipax_{max}} = \vartheta \quad \text{with} \quad \int_{t_a}^{t_b} f(t)dt = \vartheta(t_b - t_a). \tag{25}$$

Nevertheless, taking into account the few data points for an approximation with regard to Figure 14 and the commonly used approach of Eurocontrol [4], we will refer to the nine types of most used aircraft.

With respect to calculating air navigation charges, the aforementioned nine aircraft types were processed. Air navigation charges can be derived from the so called "global unit rates" which constitute fees for the en-route phase and are established by each EUROCONTROL member state, see [21]. The fees are applied on a particular sector (or airspace) and a particular MTOW of the aircraft. For the computation we refer to [21,22]. The fees also depend on the distance flown in a specific sector. The boundaries of the sectors, which

are mapped as polygons, were taken from [23]. The left hand side of Figure 15 illustrates the different sector-polygons over Europe.

Due to many possible variations in the flight plan and for the sake of completeness, the charges are estimated by using simplified great circle distances between origins and destinations, $i$ and $j$. All distances between the airports of the ICAO regions E and L were computed for the nine aircraft classes, considering the different global unit rates of the particular sectors. Due to data unavailability, particular terminal and airport charges were omitted at this step, but could be introduced in a refined approach. For further details with respect to the particular price structure of airport charges, we refer to ICAO [24].

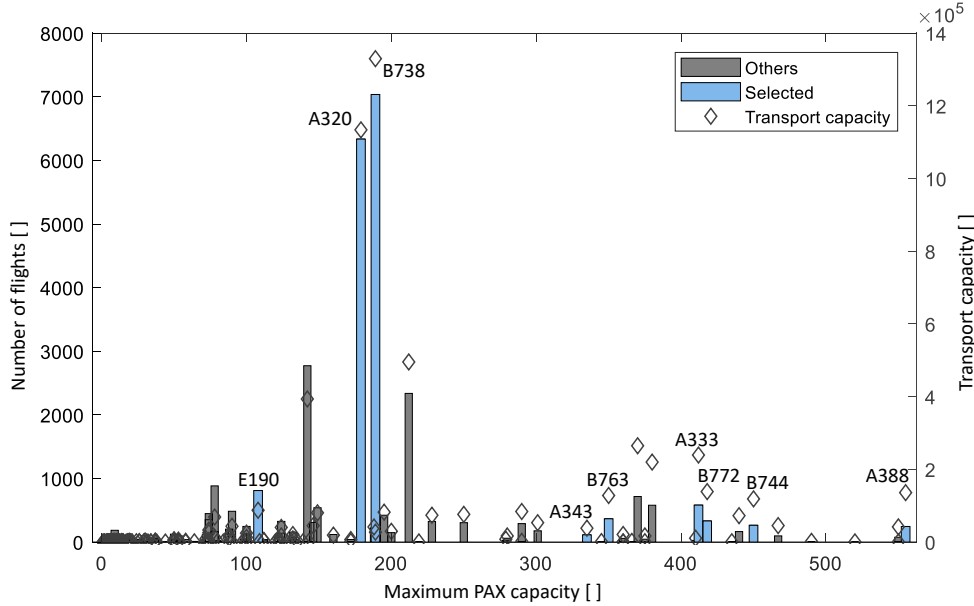

**Figure 14.** Transport capacity and number of flights over maximum passenger capacity for one selected day in Europe (complete so6 data).

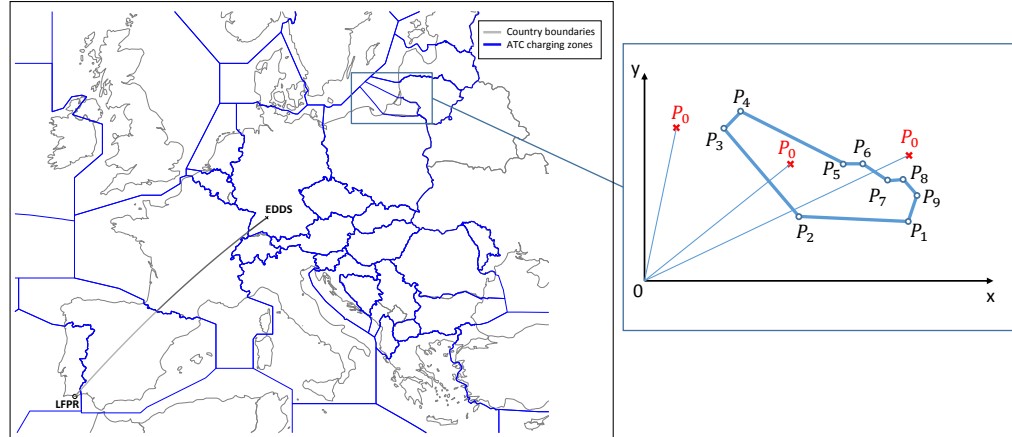

**Figure 15. Left**: Sector boundaries used for the calculation of air navigation fees on the example of EDDS and LPFR. **Right**: depiction of a polygon of a sector used to apply the Jordan point-in-polygon test.

The determination of the current sector, or polygon of which an aircraft passes through, was made by the Jordan point-in-polygon test (this method states, that for any given polygon, the number of intersections of a ray, which is directed from the outside of the polygon to any point, in- or outside the polygon, defines the relative position of the point with respect to the polygon. An odd number of intersections between ray and polygon

stands for the point being within the boundaries of the polygon. An even number locates the point outside the polygon). The right hand side of Figure 15 illustrates the method by means of one sector.

The Jordan point-in polygon procedure is applied for each particular distance $l_{ij}$ for the nine different types of aircraft $t$. In order to limit computation time, the resolution to detect in which sector an aircraft currently operates was limited to 5 km.

Thereby, the number of kilometres an aircraft covers within a specific sector, the *DistanceFactor* [km/100], is derived. The navigation charges of a sector $s$, can then be calculated by

$$charge(s) = WeightFactor(t) \cdot DistanceFactor \cdot UnitRate(country) \tag{26}$$

with

$$WeightFactor = \sqrt{MTOW/50} \tag{27}$$

and the respective global unit rate, *UnitRate* in €, for the sector, see also [21,22].

For a given distance $l_{ij}$ the particular charges are summarized and presented in https://bitbucket.org/bekir114/input-data/downloads (accessed on 24 January 2022). An excerpt of the calculated data can be found in Appendix D.

## 6. Estimation of Maintenance and Repair Costs

This chapter addresses the Maintenance, Repair and Overhaul costs of a flight, MRO. The costs for each flight leg in Equation (28) result from the block time, $time_{bl}$, and the specific cost factor, $cf_{MRO}$ [€/h], which can be derived from statistical figures, see also [25].

$$C_{MRO} = cf_{MRO} \cdot time_{bl} \tag{28}$$

$cf_{MRO}$ encompasses the following factors:

$$
\begin{aligned}
cf_{MRO} = &\left[ 13 + 4.15 \left( \frac{m_{AE}}{1000 kg} \right) \right] + \left[ 22.5 \left( \frac{P_E}{Mio.€} \right) \right] \\
&+ \left[ 0.375 \left( \frac{MTOW}{kg} Ma_{max} \right)^{0.523} \right] \\
&+ \left[ n_E \left( 9.75 + 1.875 \left( \frac{T_{max}}{10000 N} \right) \right) \right]
\end{aligned}
\tag{29}
$$

With the mass of airframe and equipment, except the engines, $m_{AE}$:

$$m_{AE} = m_{OEW} - n_E \cdot m_E \tag{30}$$

and $m_{OEW}$ being the empty mass, and $n_E$ and $m_E$ the number and mass of the engines. *MTOW* in (31) represents the maximum take-off weight, whereas $Ma_{max}$ is the maximum Mach number. The price of the engines $P_E$ adheres to:

$$P_E = c_E \cdot n_E \cdot T_{max} \tag{31}$$

where $T_{max}$ denotes the maximum thrust of an engine at sea level. The specific cost factor $c_E$ depends on $T_{max}$.

Figure 16 illustrates the relation between $c_E$ and $T_{max}$. Here, the blue area represents a corridor, which can be found in the literature, see [26]. We assumed the mean value for this corridor, which is reflected in Equation (32) .

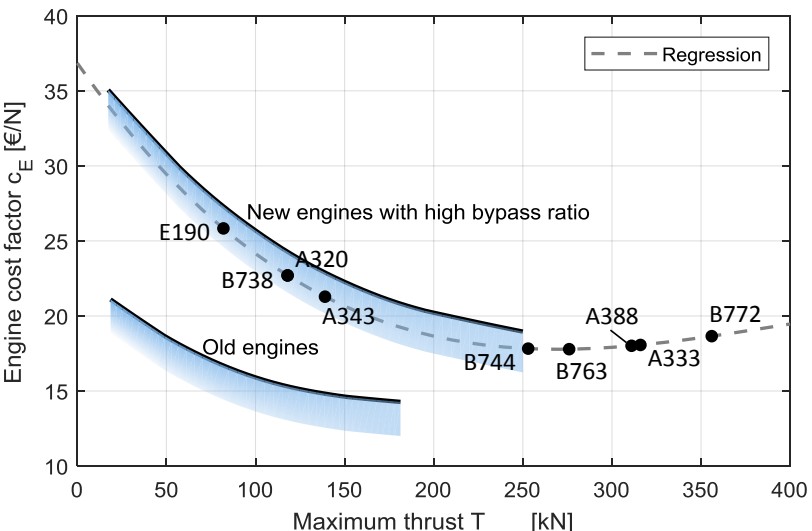

**Figure 16.** Specific engine cost factor $c_E\left[\frac{\text{€}}{N}\right]$.

The data foundation in [25] is limited to $T_{max}$ of 250 kN. In order to evaluate $c_E$ for larger engines (with a thrust of >250 kN) an extrapolation of the underlying cubic dependency was applied for larger aircraft types. The corresponding function for the engine cost factor can be stated as [26]:

$$c_E(T_{max}) = -4.0882 \times 10^{-7} T_{max}^3 + 4.8287 \times 10^{-4} T_{max}^2 \\ -0.1712\, T_{max} + 36.85 \tag{32}$$

The second term in Equation (28) equals $time_{bl} = time_{ij} + time_{taxi}$, with $time_{ij}$ and $time_{taxi}$ denoting the duration of the flight leg and the duration of the taxi time. The taxi times $time_{taxi}$ for the particular airports are currently not considered and a standard taxi time of 26 min, as in Section 4, was assumed. Though, it should be mentioned, that the taxi durations can be divided into three classes [4], depending on the capacity of airports. In a refined approach, taxi durations could be incorporated for each individual airport.

After computing the first term of Equation (28), the block time $time_{bl}$ can be derived by using Equation (21). Equation (21) represents the amount of burnt fuel for a given flown distance $l_{ij}$ in meter. The relation between $l_{ij}$ and the time an aircraft is airborne $time_{ij}$ can be expressed also by $m_{fuelburnijt}$.

$$m_{fuelburnijt} = E_{1t}time_{ij} + E_{2t}S_{ijt} + E_{3t} \tag{33}$$

Equating of (21) and (33) leads to

$$time_{ij} = \frac{D_{1t}}{E_{1t}}l_{ij} + \frac{D_{2t} - E_{2t}}{E_{1t}}S_{ijt} + \frac{D_{3t} - E_{3t}}{E_{1t}} \tag{34}$$

Since $time_{bl}$ adheres to $time_{ij} + time_{taxi}$, the assumed average duration of the taxi time (26 min), $\overline{time_{taxi}}$, is added to (34), similarly to (22), which leads to

$$C_{MRO} = cf_{MRO}(time_{ij} + \overline{time_{taxi}}) = F_{1t}l_{ij} + F_{2t}S_{ijt} + F_{3t} \tag{35}$$

Appendix E provides the MRO coefficients for the nine selected aircraft types.

## 7. Cost Estimation Functions for an All-Electric Aircraft

For the all-electric aircraft, introduced in chapter 1, an energy consumption estimation is proposed. The design concept of that aircraft provides different energy consumption rates for different phases of flight, see [1]. Those rates were applied using the same approach as in Section 4. That is, depending on 100 different routes $R_m$, with particular waypoints, climb, cruise and descending rates could be deducted. Given a constant weight of the electric aircraft, a surrogate model was implemented in order to calculate the energy consumption.

The routes were derived from the conventional aircraft type ATR-72-200 with PW124 engines, which served as a reference type in the design concept, see [1]. Due to data unavailability concerning different seat load factors, the electric aircraft was considered with a seat load factor of 100%. Figure 17 illustrates the energy consumption of the future aircraft type.

Similar to Figure 13, the regression analysis for a 100% seat load factor reveals a very good fit for a linear relationship. A quadratic relationship follows to an R² of 0.9929066 whereas the linear values follows to 0.9928974. It has to be stressed, that the applied energy consumption rates are preliminary assumptions and no real life data are available yet. On the other hand, the conventional aircraft rates of fuel flow show a good correspondence between reference values and the applied BADA implementation. A further aspect which might affect the characteristics of energy consumption is the constant mass of the electrically powered aircraft during the flight.

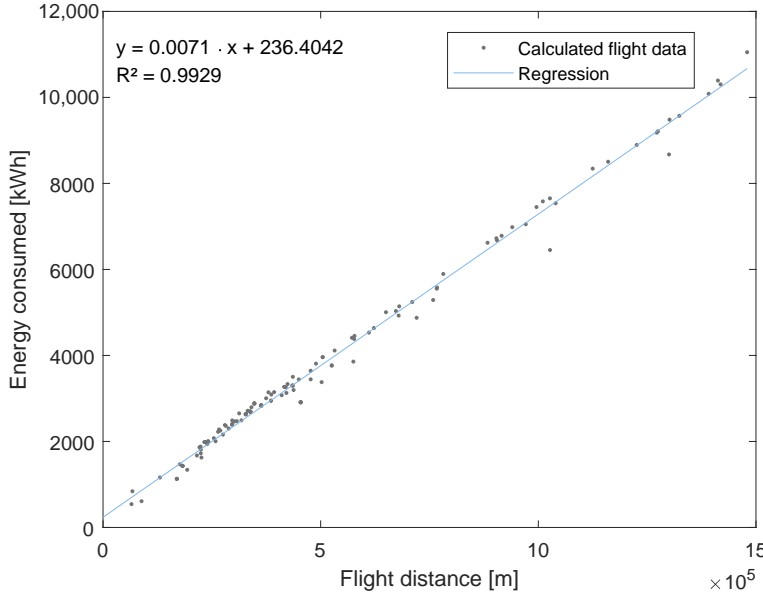

**Figure 17.** Regression analysis for the SE²A SRV1 all-electric aircraft.

The computation of air navigation charges follows the same approach as in Section 5, neglecting any possible future tax allowances for electrically powered aircraft. Concerning the estimation of the MRO costs, Equation (28) can be applied with a slight adjustment. Compared to conventional aircraft, it is asumed that the maintenance costs for the electric engines are around a third lower:

$$P_{E_{electric}} = (2/3) \cdot c_E \cdot n_E \cdot T_{max} \tag{36}$$

One of the reasons for this is that an electric motor has far fewer parts susceptible to wear and tear than an internal combustion engine, see also [27].

Other variations are expected with regard to fuel. We expect that fuel costs will increase due to raising carbon emission taxes over the last months. Due to various influencing factors we would not though forecast the development of costs for electricity. State subsidies

might decrease the price in the long run. Navigation charges might decrease as well, since emission charges will become invalid. As a counterargument holds the envisaged usage of the lower uncontrolled airspace of down to 3000 feet. This might result in higher fees from air navigation service providers, due to a higher complexity of controlling. As another aspect, which was not considered in this study, airport charges might be affected by the need for additional infrastructure and longer turnaround times due to the recharging process, optional prolonged taxi times and less noise.

Table 1 gives a summary of the SE²A SRV1 aircraft technical data and specifications. The comprehensive data set for all cost estimation functions and European airport capacities can be found under https://bitbucket.org/bekir114/input-data/downloads (accessed on 24 January 2022).

**Table 1.** Aircraft characteristics of SRV1.

| Parameter | Value | Units |
| --- | --- | --- |
| Maximum range | 800 | NM |
| Maximum number of passengers | 100 | - |
| Maximum take-off weight | 59,084 | kg |
| Operating empty weight | 49,604 | kg |
| Propulsion weight (both engines) | 1807 | kg |
| Battery weight | 24,812 | kg |
| Maximum Thrust (both engines) | 58 | kN |
| Maximum Mach number | 0.47 | - |
| Take-off field length | 1400 | m |
| Landing distance | 1100 | m |

## 8. Conclusions and Outlook

The paper discussed various general purpose cost estimation functions for a particular set of current aircraft types. In this context, it was focused on a selection of direct operating costs. The main goal was to provide means for a computational efficient calculation of emissions, which can be used to assess the trade-off between the atmospheric impact of conventional aircraft and the reduced ranges of electrically powered aircraft. For a refined approach with regard to $CO_2$ emission, the calculation of the fuel consumption was extended with regards to incorporating the passenger load. As a second cost aspect, the air navigation charges for a particular aircraft type for each pair of all origins and destinations in the European airspace were computed. In a third step an estimation function for MRO costs was presented. With respect to the introduction of new aircraft types into the European air traffic system, the energy consumption of a selected all-electric aircraft was estimated by using the same flight data set as for conventional aircraft and the particular energy consumption rates for different flight phases presented in the design concept. Estimations for navigation charges and MRO cost were proposed for the electric aircraft as well.

Depending on the current passenger demand, new short range electrically powered aircraft might change the structure of the air transportation network. The proposed cost estimation functions aim to support flow based simulation models which incorporate new all-electric short range aircraft to evaluate the performance of the new network structure. In this context, flow optimization problems, which address the introduction of new short range electric aircraft with given European wide $CO_2$ limits, can use those cost estimations. In this context, the capacity of airports is relevant. An estimation of the airport capacity, based on the runway layout was presented as well.

The derived cost estimation functions will be made publically available to enable their use in related problems as mentioned above.

**Author Contributions:** P.F. conceived the idea and performed the formal descriptions. B.Y. conceived the idea and performed the implementation of the data analysis. T.F. and P.H. gave valuable guidance and reviewed the complete paper. All authors have read and agreed to the published version of the manuscript.

**Funding:** This work has received funding from the Deutsche Forschungsgemeinschaft (DFG, German Research Foundation) under Germany's Excellence Strategy–EXC 2163/1-Sustainable and Energy Efficient Aviation–Project-ID 390881007

**Data Availability Statement:** The comprehensive data set for all cost estimation functions and European airport capacities can be found under https://bitbucket.org/bekir114/input-data/downloads (accessed on 24 January 2022).

**Acknowledgments:** We would like to acknowledge the funding by the Deutsche Forschungsgemeinschaft (DFG, German Research Foundation) under Germany's Excellence Strategy–EXC 2163/1-Sustainable and Energy Efficient Aviation–Project-ID 390881007. Furthermore, we acknowledge support by the Open Access Publication Funds of the Technische Universität Braunschweig.

**Conflicts of Interest:** The authors declare no conflict of interest. The authors declare that they have no known competing financial interests or personal relationships that could have appeared to influence the work reported in this paper.

## Abbreviations

The following abbreviations are used in this manuscript:

| | |
|---|---|
| SE²A | Sustainable and Energy-Efficient Aviation |
| SR | Short Range |
| SRV1 | SE²A SR Version 1 |
| ATS | Air Traffic System |
| ATFM | Air Traffic Flow Management |
| DOC | Direct Operating Cost |
| IOC | Indirect Operating Cost |
| SLF | Seat Load Factor |
| TAS | True Air Speed |
| BADA | Base of Aircraft Data |
| ROCD | Rate of Climb or Descent |
| ICAO | International Civil Aviation Organization |
| DRC | Declared Runway Capacity |
| RWC | Runway Configuration |
| MTOW | Maximum Take-Off Weight |
| MRO | Maintenance, Repair and Overhaul |
| MI | Mix Index |

## Appendix A. Mi and Rwc

**Table A1.** Aircraft classification for determining the airport capacity, own representation based on [13].

| Aircraft Mix Class | Aircraft Wake Turbulence Class | Number of Engines | MTOW [lb] |
|---|---|---|---|
| A | Small | Single | 12,500 or less |
| B | Small | Multiple | 12,500 or less |
| C | Large | Multiple | 12,500–300,000 |
| D | Heavy | Multiple | 300,000 or more |

**Table A2.** Runway configuration (RWC) and Mix Index (MI) for a set of selected airports.

| 1 Runway | | | 2 Runways | | | 3 Runways | | | ≥4 Runways | | |
|---|---|---|---|---|---|---|---|---|---|---|---|
| Airport | RWC | MI | Airport | RWC | MI | Airport | RWC | MI | Airport | RWC | MI |
| EBAW | A | 58 | EBLG | B | 173 | EKCH | K | 114 | EDDF | M | 146 |
| EBCI | A | 94 | EDDB | C | 96 | LIRF | L | 126 | LEMD | E | 135 |
| EBKT | A | 37 | EDDH | F | 100 | LTAI | D | 108 | LFPG | E | 155 |
| EBOS | A | 101 | EFIV | H | 98 | LTBA | M | 159 | EHAM | E | 137 |

## Appendix B. Fuel Burn Coefficients

**Table A3.** Coefficients for the calculation of fuel burn, see Equation (22): $m_{fuelburnijtaxit}[kg] = D_{1t}l_{ij}[m] + D_{2t}S_{ijt}[\%] + D_{3t} + D_{4t}time_{taxi}[min]$.

| Aircraft | D1 | D2 | D3 | D4 |
|---|---|---|---|---|
| A20N | 0.0018961 | 322.0967 | 290.7802 | 10.3329 |
| A318 | 0.0018883 | 24.5155 | 653.055 | 10.0372 |
| A319 | 0.0023529 | 79.9124 | 326.2536 | 11.5877 |
| A320 | 0.0022307 | 378.9373 | 342.0944 | 12.1564 |
| A321 | 0.0030955 | 336.251 | 71.8847 | 13.1154 |
| A332 | 0.004321 | 2336.1886 | −1003.1417 | 25.07 |
| A333 | 0.0045824 | 2984.0723 | −1068.143 | 25.4219 |
| A343 | 0.0052502 | 2779.2621 | −687.7043 | 27.2761 |
| A388 | 0.010803 | 8339.3228 | −11,351.2703 | 54.8939 |
| AT45 | 0.00082176 | 20.7657 | 71.5013 | 1.1025 |
| AT72 | 0.00072226 | 51.6356 | 131.8868 | 1.2135 |
| AT75 | 0.00090143 | 63.1776 | 76.1577 | 1.2153 |
| B712 | 0.0017305 | 150.5113 | 391.5546 | 6.7788 |
| B733 | 0.0023101 | 782.9896 | −103.9938 | 11.3237 |
| B734 | 0.0023615 | 886.2351 | 75.7369 | 12.0023 |
| B735 | 0.0020417 | 275.7148 | 323.9107 | 10.6665 |
| B737 | 0.0019398 | 494.9244 | 292.8281 | 11.6037 |
| B738 | 0.0020382 | 292.205 | 466.4805 | 11.6635 |
| B739 | 0.0022737 | 409.3189 | 226.5052 | 11.7031 |
| B744 | 0.0075432 | 5759.8071 | −3541.4171 | 45.1296 |
| B752 | 0.0027478 | 1180.4001 | −216.0449 | 15.4952 |
| B763 | 0.0043206 | 2436.3709 | −736.0163 | 26.9959 |
| B772 | 0.0052216 | 3518.65 | −3361.2039 | 26.1579 |
| B773 | 0.0054792 | 2503.1438 | −3307.2579 | 24.804 |
| B77L | 0.0055315 | 1864.8519 | −318.5394 | 35.6201 |
| B77W | 0.0062402 | 2379.0181 | −1958.6546 | 35.606 |
| BE20 | 0.00039303 | 14.6535 | 67.8146 | 0.54968 |
| BE9L | 0.00051532 | 39.7046 | 15.9969 | 0.36971 |
| C560 | 0.0003659 | 30.0301 | 113.855 | 0.5044 |
| C56X | 0.00054397 | 17.5392 | 165.985 | 2.1159 |
| C750 | 0.000466 | 367.8864 | 35.7459 | 1.4152 |
| CL60 | 0.00077628 | 94.6566 | 71.7781 | 2.1215 |
| CRJ2 | 0.00085417 | 57.2784 | 199.8621 | 3.1426 |
| CRJ9 | 0.0011746 | 68.0051 | 239.2516 | 4.957 |
| DH8D | 0.0015674 | 105.1928 | 119.575 | 2.1005 |
| E145 | 0.0010692 | 2.4672 | 205.8938 | 3.9873 |
| E170 | 0.001367 | 11.0413 | 450.1768 | 7.4721 |
| E190 | 0.0015092 | 86.7393 | 500.9485 | 8.1092 |
| F100 | 0.0017019 | 81.7194 | 371.195 | 6.0379 |

## Appendix C. Runway Capacity

**Table A4.** Estimates of hourly and annual capacities, own representation based on [13].

| Runway Configuration | | Mix index [%] | Hourly Capacity | | Annual Service Volume |
|---|---|---|---|---|---|
| | | | VFR | IFR | |
| A | | 0–20 | 98 | 59 | 230,000 |
| | | 21–50 | 74 | 57 | 195,000 |
| | | 51–80 | 63 | 56 | 205,000 |
| | | 81–120 | 55 | 53 | 210,000 |
| | | 121–180 | 51 | 50 | 240,000 |
| B | 700ft to 2499ft | 0–20 | 197 | 59 | 355,000 |
| | | 21–50 | 145 | 57 | 275,000 |
| | | 51–80 | 121 | 56 | 260,000 |
| | | 81–120 | 105 | 59 | 285,000 |
| | | 121–180 | 94 | 60 | 340,000 |
| C | 4300ft or more | 0–20 | 197 | 119 | 370,000 |
| | | 21–50 | 149 | 114 | 320,000 |
| | | 51–80 | 126 | 111 | 305,000 |
| | | 81–120 | 111 | 105 | 315,000 |
| | | 121–180 | 103 | 99 | 370,000 |
| D | 700ft to 2,499ft / 2,500ft to 3,499ft | 0–20 | 295 | 62 | 385,000 |
| | | 21–50 | 219 | 63 | 310,000 |
| | | 51–80 | 184 | 65 | 290,000 |
| | | 81–120 | 161 | 70 | 315,000 |
| | | 121–180 | 146 | 75 | 385,000 |
| E | 700ft to 2499ft / 3500ft or more / 700ft to 2499ft | 0–20 | 394 | 119 | 715,000 |
| | | 21–50 | 290 | 114 | 550,000 |
| | | 51–80 | 242 | 111 | 515,000 |
| | | 81–120 | 210 | 117 | 565,000 |
| | | 121–180 | 189 | 120 | 675,000 |
| F | | 0–20 | 98 | 59 | 230,000 |
| | | 21–50 | 77 | 57 | 200,000 |
| | | 51–80 | 77 | 56 | 215,000 |
| | | 81–120 | 76 | 59 | 225,000 |
| | | 121–180 | 72 | 60 | 265,000 |
| G | | 0–20 | 150 | 59 | 270,000 |
| | | 21–50 | 108 | 57 | 225,000 |
| | | 51–80 | 85 | 56 | 220,000 |
| | | 81–120 | 77 | 59 | 225,000 |
| | | 121–180 | 73 | 60 | 265,000 |
| H | | 0–20 | 132 | 59 | 260,000 |
| | | 21–50 | 99 | 57 | 220,000 |
| | | 51–80 | 82 | 56 | 215,000 |
| | | 81–120 | 77 | 59 | 225,000 |
| | | 121–180 | 73 | 60 | 265,000 |
| I | | 0–20 | 150 | 59 | 270,000 |
| | | 21–50 | 108 | 57 | 225,000 |
| | | 51–80 | 85 | 56 | 220,000 |
| | | 81–120 | 77 | 59 | 225,000 |
| | | 121–180 | 73 | 60 | 265,000 |

**Table A4.** *Cont.*

| Runway Configuration | Mix index [%] | Hourly Capacity | | Annual Service Volume |
|---|---|---|---|---|
| | | VFR | IFR | |
| J  | 0–20 | 132 | 59 | 260,000 |
| | 21–50 | 99 | 57 | 220,000 |
| | 51–80 | 82 | 56 | 215,000 |
| | 81–120 | 77 | 59 | 225,000 |
| | 121–180 | 73 | 60 | 265,000 |
| K  | 0–20 | 197 | 59 | 355,000 |
| | 21–50 | 145 | 57 | 275,000 |
| | 51–80 | 121 | 56 | 260,000 |
| | 81–120 | 105 | 59 | 285,000 |
| | 121–180 | 94 | 60 | 340,000 |
| L  | 0–20 | 197 | 119 | 370,000 |
| | 21–50 | 149 | 114 | 320,000 |
| | 51–80 | 126 | 111 | 305,000 |
| | 81–120 | 111 | 105 | 315,000 |
| | 121–180 | 103 | 99 | 370,000 |
| M  | 0–20 | 295 | 59 | 385,000 |
| | 21–50 | 210 | 57 | 305,000 |
| | 51–80 | 164 | 56 | 275,000 |
| | 81–120 | 146 | 59 | 300,000 |
| | 121–180 | 129 | 60 | 355,000 |
| N  | 0–20 | 295 | 59 | 385,000 |
| | 21-50 | 210 | 57 | 305,000 |
| | 51-80 | 164 | 56 | 275,000 |
| | 81-120 | 146 | 59 | 300,000 |
| | 121-180 | 129 | 60 | 355,000 |
| O  | 0–20 | 197 | 59 | 355,000 |
| | 21–50 | 147 | 57 | 275,000 |
| | 51–80 | 145 | 56 | 270,000 |
| | 81–120 | 138 | 59 | 295,000 |
| | 121–180 | 125 | 60 | 350,000 |

## Appendix D. Navigation Charges

**Table A5.** Excerpt of calculated navigation charges in € for an Airbus-A320.

| Airport | EBAW | EBBR | EBCI | EBKT | EBLG | EBMB | EBOS | EBSG | EBSH | ... |
|---|---|---|---|---|---|---|---|---|---|---|
| EBAW | 0 | 25 | 67 | 80 | 76 | 25 | 92 | 80 | 122 | |
| EBBR | 25 | 0 | 42 | 76 | 63 | 0 | 101 | 59 | 97 | |
| EBCI | 67 | 42 | 0 | 80 | 63 | 42 | 118 | 38 | 67 | |
| EBKT | 80 | 76 | 80 | 0 | 134 | 76 | 42 | 50 | 151 | |
| EBLG | 76 | 63 | 63 | 134 | 0 | 63 | 160 | 97 | 55 | |
| EBMB | 25 | 0 | 42 | 76 | 63 | 0 | 101 | 59 | 97 | |
| EBOS | 92 | 101 | 118 | 42 | 160 | 101 | 0 | 88 | 185 | |
| EBSG | 80 | 59 | 38 | 50 | 97 | 59 | 88 | 0 | 101 | |
| EBSH | 122 | 97 | 67 | 151 | 55 | 97 | 185 | 101 | 0 | |
| ⋮ | | | | | | | | | | |

## Appendix E. Mro Costs

**Table A6.** Coefficients for the calculation of MRO costs, see Equation (35).

| Aircraft | F1 | F2 | F3 |
|----------|-----------|-----------|------------|
| B738 | 0.00030509 | −2.4803 | 114.4612 |
| A320 | 0.00029891 | −2.6453 | 137.1579 |
| B772 | 0.00089164 | −14.9546 | −296.7434 |
| A333 | 0.00071983 | −4.6921 | 429.1048 |
| B744 | 0.0010803 | −64.2703 | 437.4123 |
| A343 | 0.00076361 | −4.8058 | 326.2388 |
| B763 | 0.00055659 | −7.2506 | 286.3729 |
| A388 | 0.0017044 | −344.5348 | −147.1463 |
| E190 | 0.0002165 | −1.794 | 100.1644 |

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
