# Peer review of "Approach for Cost Functions for the Use in Trade-Off Investigations Assessing the Environmental Impact of a Future Energy Efficient European Aviation"

_aerospace, doi:10.3390/aerospace9030167_

Round 1

Reviewer 1 Report

This is a good paper and addresses the surrogate model development to estimate fuel burn, navigation charge, and cost for different aircraft types, including both existing and future configurations. The paper structure is good; however, the contents require more clarity, e.g., the notations that are used in equations.

I have some comments. Please see below:

  1. The title states “approach for CO2 emission related cost functions…..” is misleading, as emission cost function could mean emission charge. But, the paper mainly addresses the fuel/energy (in terms of electric aircraft) consumption, and the cost estimation chapter 6 does not include anything like CO2 I would suggest revising the title to reveal the main focus of the paper.
  2. Figure 2 shows the current airline cost structure, which serves as an important basis for the current research in considering fuel, maintenance, and air navigation charges. However, for future electric aircraft, to what extent is this structure still valid? For instance, as the authors state the engine has fewer moveable parts hence expected lower maintenance cost, whereas the cost of other aspects will probably shift as well. Can you please elaborate on how this will affect the results of the current research?
  3. Chapter 2, can you please specify the version of BADA that is used in this paper, as different versions have different accuracy in fuel burn calculation. It might be important for the user to know what deviation is expected.
  4. Figure 3 (b), what does the color map indicate?
  5. Figure 5 shows the traffic flow among different seasons. But later on, for the regression, it seems that the seasons are not considered at all. Is this correct? Or perhaps I overlooked it.
  6. Line 307: “…a optimization model…” -> “… an optimization model…”
  7. Line 340: “as mentioned above, for each Sijt is represented by 100 different routes” ->incomplete sentence
  8. Line 342: B -> Appendix B.
  9. Figure 10. What is SLF and its unit?
  10. Line 359: TCt appears for the first time. Please provide the full name.
  11. Line 360: Please add an explanation on tpaxmax?
  12. Equation (29) comprises different shares of cost. Can you explain what does each term in this equation stands for? i.e., which cost are they?
  13. Figure 13. The regression curve seems to be increasing for maximum thrust beyond 250 kN. Can you please provide explanations?
  14. Lines above equation 31: “MTOW in (31)” -> (29)?; “maximum take of weight “->maximum takeoff weight
  15. The model has been presented in the paper, but somehow, no validations are presented. This should be addressed for publications.
  16. The current paper considers no weather, meaning no wind effects, which usually affect fuel burns. I understand addressing this might be out of the scope of this paper. However, can you please add discussions of the expected influences on the models?

Reviewer 2 Report

General comments

The authors have submitted a very interesting paper which presents a novel approach to calculating the environmental impact of ATM operations. This approach would hold considerable value to various industry stakeholders given the low computational expense that the authors have focussed on.

As reviewer, my primary concern is the lack of validation of the developed algorithms and coefficients. In order for this approach to be adopted with confidence, there needs to be validation evidence provided (although the assumptions are sound). This could be done per section, or as a separate section validating all aspects in combination. I would recommend that validation be conducted either against recorded flight data for the flight cases considered, or against other commonly used environmental impact methodologies.

The paper does stand-up as it is written, but with reduced impact.

Best wishes with your future research.

Other specific comments are below:

Line 48: ‘exemplary’ means “serving as a desirable model; very good”, do the authors just mean ‘example’?

Lines 129 to 144: There is considerable variation in how aircraft actually fly their filed planned route, such that the turn trajectory calculations feels unnecessary for the level of aggregation that the cost functions are ultimately being presented at. The paper would benefit from some justification regarding why this approach was deemed necessary.

Section 3 shows an interesting, novel approach to calculating airport capacity however there is no validation of this approach. The paper would benefit from a simple regression analysis to show how the algorithm-estimated airport capacity compares to the 5% peak hour movements.

Line 253: do you mean 300 different types of aircraft? Or 300 different combinations when accounting for all of the other factors?

Line 301: Might resemble concerning – is there a word missing?

Figure 10: I am surprised to see a linear relationship. It’s widely acknowledged that shorter flight legs consume more fuel due to the airport / climb phases of flight being more fuel intensive. Are the constants of D3t + D4t timetaxi being considered here?

Line 362 – 363: missing line numbering. Also before Eq. 25 should be ‘of passengers of a new class’

Figure 14: As with Fig 10, I’m a little surprised to see a linear relationship. It may be worth some text to explain this.

Round 2

Reviewer 1 Report

Thank you for the revision.